# The Roles of Long Noncoding RNAs HNF1α-AS1 and HNF4α-AS1 in Drug Metabolism and Human Diseases

**DOI:** 10.3390/ncrna6020024

**Published:** 2020-06-24

**Authors:** Liming Chen, Yifan Bao, Suzhen Jiang, Xiao-bo Zhong

**Affiliations:** 1Department of Pharmaceutical Sciences, School of Pharmacy, University of Connecticut, Storrs, CT 06269, USA; liming.chen@uconn.edu (L.C.); yifan.bao@uconn.edu (Y.B.); suzhen.jiang@uconn.edu (S.J.); 2School of Pharmaceutical Sciences, Guangzhou University of Chinese Medicine, Guangzhou, Guangdong 51006, China

**Keywords:** lncRNA, *HNF1α-AS1*, *HNF4α-AS1*, cancer, cytochrome P450

## Abstract

Long noncoding RNAs (lncRNAs) are RNAs with a length of over 200 nucleotides that do not have protein-coding abilities. Recent studies suggest that lncRNAs are highly involved in physiological functions and diseases. lncRNAs HNF1α-AS1 and HNF4α-AS1 are transcripts of lncRNA genes *HNF1α-AS1* and *HNF4α-AS1*, which are antisense lncRNA genes located in the neighborhood regions of the transcription factor (TF) genes *HNF1α* and *HNF4α*, respectively. HNF1α-AS1 and HNF4α-AS1 have been reported to be involved in several important functions in human physiological activities and diseases. In the liver, HNF1α-AS1 and HNF4α-AS1 regulate the expression and function of several drug-metabolizing cytochrome P450 (P450) enzymes, which also further impact P450-mediated drug metabolism and drug toxicity. In addition, HNF1α-AS1 and HNF4α-AS1 also play important roles in the tumorigenesis, progression, invasion, and treatment outcome of several cancers. Through interacting with different molecules, including miRNAs and proteins, HNF1α-AS1 and HNF4α-AS1 can regulate their target genes in several different mechanisms including miRNA sponge, decoy, or scaffold. The purpose of the current review is to summarize the identified functions and mechanisms of HNF1α-AS1 and HNF4α-AS1 and to discuss the future directions of research of these two lncRNAs.

## 1. Introduction

The advancement of next-generation sequencing techniques makes it possible to investigate the human genome in a clearer and more detailed aspect. In addition to the coding genes, the noncoding genes, which actually take up the majority of the human genome, have also been investigated [1]. The transcripts of noncoding genes are defined as noncoding RNAs (ncRNAs). Initially, ncRNAs were believed to have no protein coding function as they do not have protein-coding open-reading frames (ORFs). However, this idea was challenged by several recent findings, showing that some ncRNAs can generate microproteins/small peptides, which gave rise to a new insight into the ncRNA functions [2,3]. Along with the progress in the identification of all kinds of ncRNAs, the studies focusing on the functions and mechanisms of ncRNAs are also boosting. 

Long ncRNAs (lncRNAs) are a group of ncRNAs with more than 200 nt in length [4]. lncRNAs are found to be highly involved in the regulation of gene expression and functions. This feature of lncRNAs makes them also important for many biological and physiological processes in human and other organisms. Up to date, several lncRNA databases have been established to assist the study of lncRNAs, including the GENCODE, LNCipedia, ENCODE, and NONCODE [5,6,7,8]. According to the GENCODE database, 17,960 lncRNAs have been annotation in the human genome in 2020 [5]. However, compared to the large number of annotated lncRNAs, far fewer lncRNAs have been functionally characterized and understood. A huge knowledge gap remains in understanding how lncRNAs can impact on human health.

Hepatocyte nuclear factor 1 alpha-antisense-1 (HNF1α-AS1) and HNF4α-antisense-1 (HNF4α-AS1) are two recently identified lncRNAs in human. They are named based on their genomic locations, where the noncoding genes *HNF1α-AS1* and *HNF4α-AS1* are located next to two important transcription factor (TF) genes *HNF1α* and *HNF4α* and are transcribed in the opposite direction on the antisense strand. Several special functions and mechanisms of these two lncRNAs have been found in normal physiologic activities and abnormal diseases. In the current article, the major findings of HNF1α-AS1 and HNF4α-AS1 are summarized and the future directions of studies about these two lncRNAs are discussed. 

## 2. Neighborhood Antisense lncRNAs HNF1α-AS1 and HNF4α-AS1

### 2.1. Classification of lncRNAs Based on Genomic Locations

lncRNA genes can be found in several different locations relative to coding genes in the genomes (Figure 1) [9]. Intergenic lncRNAs (lincRNAs) are lncRNAs located between two coding genes. Intronic lncRNAs are lncRNAs positioned within the intronic region of a coding gene. Sense lncRNAs are lncRNAs transcribed from the same strand of a coding gene, while antisense lncRNAs are lncRNAs transcribed from the opposite strand of a coding gene. 

lncRNAs can also be divided into *cis*-acting or *trans*-acting lncRNAs based on their functional type [10,11]. lncRNAs acting as *cis*-regulators are mainly involved in the expression and function of their neighborhood coding genes, which are also likely to show expression correlation with coding genes. Examples for *cis*-acting lncRNAs include lncRNA VIM-AS1 and CARMEN [12,13]. On the other hand, *trans*-regulatory lncRNAs are involved in the regulation of distal genes from their transcription sites, where lncRNA FIRRE is an example [14]. However, with the progression in understanding lncRNA function, lncRNAs with both *cis*- and *trans*-regulatory functions have been identified, where the lncRNA lncKdm2b is an example [15,16].

### 2.2. Neighborhood Antisense lncRNAs to Sense Genes

An antisense transcript refers to a transcript generated from the opposite strand to a protein-coding or sense strand [17]. The existence of antisense transcription is very common across different transcriptomes and these antisense transcripts are usually co-transcribed with their sense transcripts [18]. It has been demonstrated that more than 60% of the sense transcripts have antisense partners [17,19,20]. Antisense transcripts have been reported to have multiple functions in the regulation of gene expression, which have further impacts on human health and diseases [21,22]. 

lncRNAs have been identified to be one common type of antisense transcripts of coding genes and are involved in the regulation of function and expression of their neighborhood coding genes [23,24]. These antisense lncRNAs also have several special characteristics, which are similar to mRNAs. There are several structural similarities between the lncRNAs and mRNAs, where both can be multi-exonic, 5′-capped, and 3′-polyadenylated [25]. The processing of lncRNAs is also similar to mRNAs, where the RNA polymerase II is responsible for the synthesis and both RNAs need to be spliced [21]. The transcriptional activity of lncRNAs can also be regulated by special DNA elements in promoters and enhancers, where promoters can have different locations [26]. In some special cases, both the sense and antisense genes can be controlled by a same bidirectional promoter, which causes the co-expression of the sense and antisense gene pair [27].

There are several locations near a coding gene where antisense lncRNAs are likely to exist (Figure 2). However, it is still not clear how different locations affect the function of lncRNAs. Recent studies have found that these neighborhood lncRNAs might affect their nearby coding genes in multiple ways, including direct regulation of the coding genes or indirect involvement in the functional regulation of the coding genes. 

### 2.3. Neighborhood Antisense lncRNAs to TFs

Along with the advancements in the discovery of lncRNAs, there is an increasing amount of studies trying to identify lncRNAs that have special relationships with coding genes. These relationships include but are not limited to correlations in expression, spatial localization, and function [28,29,30]. There is increasing evidence showing that lncRNAs, including TF neighborhood antisense lncRNAs, are involved in the expression or functions of TFs in gene regulation [31]. In this section, several TF-lncRNA pairs are summarized and discussed (Table 1).

*GATA3-AS1* is a divergent lncRNA gene sharing a promoter region with the TF gene *GATA3*, encoding a master regulator in T helper 2 cell functions. In one study performed by Gibbons et al., GATA3-AS1 was found to control the expression of *GATA3* gene as well as two interleukin genes, *IL5* and *IL13*. Specifically, depletion of GATA3-AS1 resulted in a reduction in GATA3, IL5, and IL13 mRNA and protein levels, along with decreased T helper 2 cell polarization. This regulatory function of GATA3-AS1 was accomplished through the interaction with histone-modifying enzymes and the alteration of histone codes to its own gene locus, which is the shared promoter region of *GATA3* [32]. 

GA-binding protein subunit beta-1 (GABPB1) is a TF known to regulate the transcription of various genes related to antioxidation process, including Peroxiredoxin 5 (*PRDX5*) [35]. lncRNA *GABPB1-AS1* gene is an antisense RNA gene of the *GABPB1* gene. GABPB1 protein was previously reported to respond to chemical-induced cellular stress [36]. A recent study showed that GABPB1-AS1 was able to affect the functions of GABPB1 by modulating the translation process of GABPB1 in HepG2 cells, an *in vitro* model for hepatocellular carcinoma (HCC). The upregulation of GABPB1-AS1 inhibited the translation of GABPB1, which further reduced the expression of peroxidase genes, including PRDX5, and accumulation of reactive oxygen species and lipid peroxidation in the cells [33]. 

Yin Yang 1 (YY1) is an ubiquitously expressed TF involved in the regulation of cell proliferation and differentiation [37]. In muscle cells or myoblasts, YY1 regulates genes involved in cell proliferation and differentiation by interacting with histone methyltransferase complex and altering histone markers on target genes [38]. *Linc-YY1* gene is a divergently transcribed lncRNA gene located upstream of the *YY1* coding gene. The employment of linc-YY1 loss- and gain-of-function assays in a mouse myoblast cell line showed that linc-YY1 was important in the myogenic differentiation and muscle regeneration processes, and modulation of *linc-YY1* also affected YY1 target genes. This function of linc-YY1 was accomplished through interacting with YY1 directly and eviction of YY1/Polycomb repressive complex, a complex mediating histone methylation from promoter regions of target genes [34].

All these studies supported the idea that lncRNAs, especially neighborhood lncRNAs of TFs, might be involved in the function of these TFs. 

### 2.4. Genomic Locations and Structures of HNF1α-AS1 and HNF4α-AS1

*HNF1α-AS1*, a neighbor antisense lncRNA gene of the human *HNF1α* gene, is located at human Chromosome 12 with a length of 39.04 kb, containing two exons and one intron. The coding gene *HNF1α* and noncoding gene *HNF1α-AS1* formed a typical TF-lncRNA pair (Figure 3A). 

*HNF4α-AS1*, a neighbor antisense lncRNA gene of the human *HNF4α* gene, is located at human Chromosome 20 with a length of 17.96 kb, containing four exons and three introns. The *HNF4α* and *HNF4α-AS1* genes also form a typical pair of coding and neighborhood antisense noncoding genes (Figure 3B). 

The coding potentials of HNF1α and HNF1α-AS1 as well as HNF4α and HNF4α-AS1 were analyzed by the Coding Potential Calculator (http://cpc2.cbi.pku.edu.cn) [39]. The analysis results (Table 2) indicate that both HNF1α-AS1 and HNF4α-AS1 have low coding probabilities in comparison to HNF1α and HNF4α, which confirmed the noncoding features of HNF1α-AS1 and HNF4α-AS1.

The RNA transcripts of both coding genes of *HNF1α* and *HNF4α* and noncoding genes of *HNF1α-AS1* and *HNF4α-AS1* show tissue-specific distributions in normal human tissues. To compare the expression patterns between the coding and noncoding genes, the top 10 tissues expressing HNF1α, HNF1α-AS1, HNF4α, and HNF4α-AS1 in human are summarized and compared. The tissue distribution pattern of HNF1α mRNA was retrieved from the RNA-Seq Expression Data GTEx in 53 tissues from 570 donors and shows relatively higher expression levels in the stomach, small intestine, colon, liver, pancreas, and kidney, the major organs in the gastrointestinal (GI) tract (Figure 4) [40]. A similar pattern in the GI tract organs is also found for HNF1α-AS1 (Figure 4A), indicating that HNF1α-AS1 is expressed in the organs where HNF1α is expressed. These results may suggest that HNF1α-AS1 is possibly involved in the regulatory function of HNF1α in the GI tract organs. Similar tissue distribution patterns are also found between HNF4α and HNF4α-AS1 (Figure 4B). These similarities in tissue distribution might suggest functional connections between HNF1α and HNF1α-AS1 as well as between HNF4α and HNF4α-AS1. 

In addition to the chromatin locations and tissue distribution, the structures of lncRNAs are also important for their function and mechanism of action. RNA molecules have been shown to adopt higher-order tertiary interactions [41]. Even though the relationships between lncRNA structure and functions are still not fully understood yet, identification of structural domains, which mediate the interactions between lncRNAs and other molecules, is critical for the characterization of lncRNA functions. The secondary structures of HNF1α-AS1 is predicated using RNAfold program (http://rna.tbi.univie.ac.at//cgi-bin/RNAWebSuite/RNAfold.cgi). The secondary structure of HNF4α-AS1 is not predicated as the sequence length of HNF4α-AS1 exceeds the upper limit of lncRNA FASTA input for the program. 

As shown in Figure 5, HNF1α-AS1 is able to form a stable secondary structure based on minimum free energy calculation. Several domains are also observed on the secondary structure of HNF1α-AS1. However, it is still not very clear what molecules interact with these domains and it is still largely unknown how they impact on the functions of HNF1α-AS1. 

HNF1α and HNF4α are well-studied TFs involved in organ maturation, cell differentiation, and disease development. Both HNF1α and HNF4α can be detected during different stages of embryonic development and they participate in the development of multiple organs, including liver, colon, and pancreas [42,43,44]. HNF1α and HNF4α are also regarded as master regulators of the metabolic functions in human. The target genes of HNF1α and HNF4α are involved in lipid metabolism, bile acid synthesis, lipoprotein metabolism, glucose metabolism, amino acid metabolism, and xenobiotic metabolism [45,46,47,48,49]. Taking xenobiotic metabolism as an example, knockdown of HNF1α or HNF4α in mice or human primary hepatocytes led to downregulation of the mRNA expression of several cytochrome P450s (P450s), which play critical roles in metabolizing drugs [49,50]. In addition to normal physiological activities, HNF1α and HNF4α are also highly involved in several diseases. Genetic mutations in *HNF1α* or *HNF4α* gene are some of the most common causes of maturity-onset diabetes of the young (MODY), which is characterized by a non-insulin dependent form of diabetes in young people below the age of 25 [51]. The dysfunction of pancreatic beta cells and impaired insulin secretion because of HNF1α or HNF4α mutations are believed to be the pathological mechanisms to this type of disease. In addition to MODY, HNF1α and HNF4α are involved in other type of diseases, including other metabolic diseases, inflammatory diseases, and cancer [52,53,54,55]. HNF1α and HNF4α are reported to play different roles based on cancer types. In HCC, the transduction of HNF1α and HNF4α was showed to suppress the growth of HepG2 and Huh7 cells *in vitro* and to reduce the tumorigenicity of these cells after transplantation into mice, indicating a role of tumor suppressing of these TFs [56]. However, in pancreatic cancer, the role of HNF1α is controversial based on the results from two reports. Luo and his colleagues reported that HNF1α was a possible tumor suppressing gene in pancreatic cancer [57]. The results of immunohistochemistry showed that the level of HNF1α was significantly lower in pancreatic cancer tissues than normal pancreatic tissues. Furthermore, knockdown of HNF1α also led to increase proliferation and to decrease apoptosis in pancreatic cancer cell lines, which supported the conclusion that HNF1α plays a role in tumor suppressing. This conclusion was challenged by another report by Abel et al., in which the authors showed that overexpression of HNF1α increased the formation of pancreatic cancer stem cells and tumorsphere [58]. In summary, HNF1α and HNF4α are two very important TFs in human physiologic functions and diseases.

Comparing to the well identified functions of HNF1α and HNF4α, the underlying molecular mechanisms of how HNF1α and HNF4α perform these functions is still not clearly understood yet. As TFs in nature, HNF1α and HNF4α can regulate their target genes through directly binding to their promoter regions. Other than direct binding, some other mechanisms of HNF1α and HNF4α in gene regulation are still elusive and need to be identified. lncRNAs are found to serve as cofactors of several important regulatory proteins and to be involved in their functions. However, whether lncRNAs are also involved in the function of HNF1α and HNF4α is still not fully understood. 

## 3. Relations of the Regulation of Expression between HNF1α and HNF1α-AS1 and between HNF4α and HNF4α-AS1

The knowledge regarding how the expression of HNF1α-AS1 and HNF4α-AS1 is regulated in cells is still very limited. According to the literature, these lncRNAs are regulated by their neighborhood gene-coded proteins, which are HNF1α and HNF4α.

One study identified HNF1α-AS1 as an HNF1α-regulated lncRNA by comparing expression of HNF1α-AS1 in Huh7 cells with overexpression or knockdown of HNF1α. This result was also confirmed by the existence of HNF1α response element in the promoter region of HNF1α-AS1 and the positive expression correlation between HNF1α and HNF1α-AS1 in several HCC cell lines and human HCC samples [59]. In addition, knockdown of HNF1α led to the depletion of HNF1α-AS1 RNA levels in Huh7 or HepaRG cells, which further indicated this regulation relationship [60,61]. Other proteins were also reported to regulate the expression of HNF1α-AS1. Early growth response protein 1, a TF involved in cancer development, was also reported to regulate expression of HNF1α-AS1 by directly binding to the promoter region of HNF1α-AS1 and activating its transcription [62]. 

Guo and Lu reported that the expression of HNF4α-AS1 was strongly activated by P1-HNF4α, which is predominantly produced in adult liver, but not P2-HNF4α, which is prevalent in fetal liver, pancreas, and liver/colon cancer. Therefore, HNF4α-AS1 might be a biomarker for P1-HNF4α expression and might be involved in the functional regulation of the liver-specific P1-HNF4α [63]. The knockdown of HNF4α also showed to repress the expression of HNF4α-AS1 in HepaRG cells, which further validated the idea the HNF4α-AS1 was regulated by HNF4α [60].

These studies suggested the expression regulatory hierarchy between these two TF-lncRNA pairs, where the expression of lncRNAs, HNF1α-AS1 and HNF4α-AS1, are regulated by their neighborhood TFs, HNF1α and HNF4α. However, there is still no clear evidence showing the direct binding of HNF1α to the HNF1α-AS1 promoter or HNF4α to the HNF4α-AS1 promoter. Whether HNF1α and HNF4α regulate HNF1α-AS1 and HNF4α-AS1 through direct binding or there are still other regulatory mechanisms between the TFs and lncRNAs are still not understood. Future studies would be needed to address these questions.

## 4. Functions of HNF1α-AS1 and HNF4α-AS1 in Human Physiology and Diseases

### 4.1. lncRNAs in Human Physiology and Diseases

lncRNAs have been found to highly involved in both normal physiological activities and diseases in human health [64,65].

Numerous lncRNAs have been found to play critical roles in the processes of development, including organ development and embryogenesis. LncRNA FENDRR, a divergent lncRNA of TF FOXF1, was one of the examples reported to be involved in organ development. Knockdown of FENDRR in mice led to defects in multiple organs, including the heart, lung, and gastrointestinal tract [66]. Another example lncRNA, NEAT1, was reported to be involved in the development of mammary glands and pregnancy in mice [67]. Other than development, lncRNAs have been found in other physiological processes, such as energy metabolism, circadian rhythm, and spermatogenesis [68,69,70]. 

Diseases related lncRNAs are extensively studied recently. Utilizing the sequencing techniques, thousands of differentially expressed lncRNAs with potential implications in the initiation, progression, and treatment of all kinds of diseases have been identified and studied. In a comprehensive study comparing the transcriptome among more than 7000 samples, including tumor tissues, normal tissues, and tumor cell lines, thousands of cancer-related lncRNAs have been identified [71]. This study opened up the research of using lncRNAs as biomarkers for cancer types, progression, or treatment strategies. In recent years, more and more cancer-related lncRNAs have been identified and functionally characterized [72]. Aside from cancer, lncRNAs are also important for other diseases. For example, lncRNA BACE1-AS is reported to have a strong correlation with Alzheimer’s disease and lncRNA MHRT is involved in cardiovascular diseases [73,74]. 

### 4.2. Roles of HNF1α-AS1 and HNF4α-AS1 in Physiologic Functions, including Drug Metabolism

HNF1α-AS1 and HNF4α-AS1 were recently reported to be involved in the hepatic metabolic function and drug metabolism through affecting P450 genes and related TFs [60,61,75]. 

Using a lncRNA specific microarray assay, the differences of lncRNA expression were compared between human liver samples with high and low CYP3A4 expression. HNF1α-AS1 was identified as one of the differentially expressed lncRNAs [61]. In addition, positive correlations were identified in human liver samples between the expression of HNF1α-AS1 and CYP2C8, 2C9, 2C19, 2D6, 2E1, 3A4, pregnane X receptor (PXR), constitutive androstane receptor (CAR), and HNF1α, which implicated the involvement of HNF1α-AS1 in regulating drug metabolizing enzymes (DMEs) and related TFs in the liver. The functions of HNF1α-AS1 was further studied by generating loss-of-function models in HepaRG and Huh7cell lines. In both cell lines, knockdown of HNF1α-AS1 led to decreases in mRNA expression of several major P450s involved in drug metabolism, as well as P450 related TFs [60,61]. Interestingly, HNF4α-AS1 was reported to have opposite regulatory effects comparing to HNF1α-AS1. Specifically, knockdown of HNF4α-AS1 in HepaRG cells resulted in induction of several P450s, including CYP1A2, 2B6, 2C8, 2C9, 2C19, 2E1, and 3A4, as well as P450 related TFs PXR and CAR [60]. 

In a follow-up study performed by Chen and his colleagues, knockdown of HNF1α-AS1 or HNF4α-AS1 also caused alterations in the susceptibility of acetaminophen (APAP)-induced cytotoxicity in HepaRG cells [75]. The metabolic pathway analysis showed that the alterations of toxicity were primarily caused by the changes in APAP-metabolizing P450s in both mRNA and protein levels. These reported physiological roles of HNF1α-AS1 and HNF4α-AS1 are summarized and displayed in Figure 6.

These results indicate that the lncRNAs HNF1α-AS1 and HNF4α-AS1 are involved in the normal physiological functions of the liver by regulating the metabolic functions. Recent studies have suggested that lncRNAs are highly relevant to drug metabolism through the regulation of drug metabolizing enzyme genes [76,77,78]. In addition to HNF1α-AS1 and HNF4α-AS1, several other lncRNAs have also been identified in the regulation of expression of drug metabolizing enzyme genes, such as LINC00574 in the regulation of UGT2B15 expression, and contribution to susceptibility of drug-induced liver injury (DILI), for example LINC00844 [79,80]. These works have opened a novel area in the field of drug metabolism and DILI, which requires further studies to provide answers for some fundamental questions.

### 4.3. Role of HNF1α-AS1 and HNF4α-AS1 in the Progress of Human Diseases, including Cancers

Several functions of HNF1α-AS1 and HNF4α-AS1 in human diseases have been reported in publications. In this section, the identified functions of HNF1α-AS1 and HNF4α-AS1 in human diseases are summarized and discussed.

Currently, numerous studies have shown that HNF1α-AS1 is actively involved in different stages of cancer, including tumorigenesis, progression, and treatment. Significantly differentially expressed HNF1α-AS1 has been found in several types of cancer tissues comparing to normal tissues, including the liver, colon, lung, cervical, stomach, and bladder. However, the regulation patterns are different. Specifically, HNF1α-AS1 is upregulated in esophageal, lung, bladder, and colon cancers and downregulated in gastric and pancreatic cancers [81,82,83,84,85,86]. These results indicate that HNF1α-AS1 might have tissue-specific functions. Most studies showed that the high expression levels of HNF1α-AS1 were associated with a poor prognosis, a higher risk of metastasis, and a lower overall survival rate in cancers [82,85,87,88,89]. 

HNF1α-AS1 plays a contradictory role in the development of cancer. According to current studies, HNF1α-AS1 was reported to function as both oncogene and tumor-suppressing gene in different cancers. Zhang et al. reported that HNF1α-AS1 promoted carcinogenesis in colorectal cancer by activating the Wnt/β-catenin signaling pathway, indicating that HNF1α-AS1 might be used as a prognostic biomarker in colorectal cancer [90]. In HCC, HNF1α-AS1 was reported to work as a tumor repressor by decreasing tumor growth and metastasis [59]. However, two other studies focusing on the roles of HNF1α-AS1 in HCC defined HNF1α-AS1 as an oncogene based on its function to promote cell proliferation and hepatocarcinogenesis [89,91].

For the progression of cancer, HNF1α-AS1 mainly regulates tumor growth and metastasis. Knockdown of HNF1α-AS1 was reported to reduce cell proliferation rates in several *in vitro* cancer cell lines, which might further affect tumor growth [88,92,93]. Metastasis is another major activity during cancer progression. HNF1α-AS1 was also reported to be involved in cancer metastasis by regulating cell migration, invasion, colony formation, and epithelial–mesenchymal transition (EMT), and knockdown of HNF1α-AS1 resulted in decreased cell movement or invasion, which might further impact the metastasis of tumor [85,88,92,93,94,95]. In this part of function, HNF1α-AS1 was involved in the several signaling pathways to promote cell survival and movement, including the CDK signaling pathway, p53 pathway, and the Akt-mTOR/GSK3β signaling pathway. 

Drug resistance is the major cause of treatment failure in cancer treatment. Several mechanisms have been identified to contribute to the development of drug resistance in cancer cells, including promoted cell survival and altered drug metabolism [96]. The expression of HNF1α-AS1 was reported to affect the sensitivity of cancer cells to anti-cancer drugs. Cisplatin is a widely used chemotherapeutic drug for the treatment of cervical cancer. In one study performed by Luo et al., knockdown of HNF1α-AS1 in cisplatin-resistant HeLa re-sensitized the cells to cisplatin treatment, indicating that HNF1α-AS1 was involved in the regulation of drug resistance in cancer cells [97].

In summary, based on the findings in the current published articles, HNF1α-AS1 is an important regulatory molecule in cancer biology. However, controversy still exists in the roles of HNF1α-AS1 in cancers and future studies are needed to clarify the functions of HNF1α-AS1.

In terms of HNF4α-AS1, the functional study is still very limited. HNF4α-AS1 was found to be differentially expressed in HCC compared with adjacent cirrhotic tissues using RNA sequencing techniques, indicating that HNF4α-AS1 may function as a biomarker for the development of HCC in patients with cirrhosis [98]. Using co-expression and functional annotation enrichment analyses across several tissues and cell types, Haberman and his colleagues reported that HNF4α-AS1 was significantly downregulated in patients with Crohn’s disease, where the expression of HNF4α-AS1 also showed association with an epithelial metabolic signature, indicating that HNF4α-AS1 might be a novel inflammatory signal or therapeutic target for Crohn’s disease [99].

## 5. Potential Mechanisms of HNF1α-AS1 and HNF4α-AS1 in the Regulation of Gene Expression

### 5.1. lncRNAs in Gene Regulation: Functions and Mechanisms

The “central dogma of molecular biology” describes the flow of genetic information, where the genetic information coded by DNAs is transcribed into mRNAs and mRNAs are further translated into proteins. lncRNAs have been reported to be involved in nearly every steps of gene regulation, including transcription and translation processes.

The mechanisms of how lncRNAs regulate gene expression are complex and have not been fully understood yet. Comparing to the identified amount of lncRNAs, there are far fewer lncRNAs that have been functionally defined and assigned with explicit roles. Nevertheless, these lncRNAs have shown their significance and importance in the regulation of gene expression [9,100]. Current insights into the lncRNAs suggest that lncRNAs can interact with different biological molecules, including RNAs [101,102], DNAs [101,103,104], and proteins [105,106] (Figure 7). The interactions with DNAs and formation of lncRNA–DNA duplexes or triplexes are the basis of lncRNAs in gene regulation at the pre-transcriptional level. Examples of lncRNAs interacting with DNAs include ANRASSF1 and ANRIL [107,108]. lncRNAs interacting with RNAs usually associate with gene regulation at the post-transcriptional level, which modulates mRNA expression and functions. Examples of lncRNAs which are able to interact with RNAs include BACE1-AS1 and PTENpg1 [109,110,111]. lncRNAs interacting with proteins can perform diverse functions based on the protein functions and are involved in all levels of gene regulation. Xist, APOA1-AS, and HOTAIR are example lncRNAs that perform their functions through interacting with proteins [112,113,114].

By interacting with other molecules, lncRNAs can perform their regulatory functions through four defined molecular mechanisms: decoy, guide, scaffold, and signal (Figure 8) [115]. lncRNAs function as decoys, which can preclude the interactions between molecules, such as DNA–protein interactions. Examples for this type of lncRNAs are Gas5 and PANDA [116,117]. LncRNAs functioning through guide and scaffold mechanisms share similar features. By binding with multiple molecules and concentrating them in certain sub-cellular areas, the lncRNAs can increase the interaction between these recruited molecules, including protein–protein interaction, protein–DNA interaction, and protein–RNA interaction. One of the well-studied scaffold lncRNA examples is HOTAIR, which can bind with two histone modification complexes, PRC2 and LSD1, and is involved in gene regulation by histone tail modification [118]. Signal lncRNAs are defined by their function to activate or to repress genes or their ability to change the chromatin conformation. Enhancer lncRNAs are reported to have this signal function in gene regulation, with lncRNA HOTTIP as an example. HOTTIP was reported to directly interact with WDR5, a key component protein in histone modification, inducing the formation of chromosome looping and enforcing gene activation by maintaining histone H3K4me3 modification [119]. However, this archetype of classification is oversimplified sometimes as one lncRNA may function through multiple mechanisms. One example is the lncRNA Xist, which can regulate its target genes through both signal and guide mechanisms [120,121]. 

### 5.2. Mechanisms of HNF1α-AS1 in Gene Regulation

Identifying the interacting molecules to lncRNAs is the key to understanding the mechanisms involved in lncRNA functions. According to current literature, HNF1α-AS1 functions by interaction with miRNAs and proteins. However, what are the interacting molecules to HNF4α-AS1 is still largely unknown.

HNF1α-AS1 has been reported to interact with miRNAs in cancer cells to regulate cell characteristics, including proliferation, movement, and apoptosis. By acting as a miRNA sponge, HNF1α-AS1 was able to dis-inhibit the expression of genes by sponging miRNAs away from their target genes. The miRNAs interacting with HNF1α-AS1 are summarized in Table 3. 

HNF1α-AS1 was also reported to interact with functional proteins involved in cancer. Ding et al. found that HNF1α-AS1 was able to directly bind to the C-terminal of Src homology region 2 domain-containing phosphatase 1 (SHP-1) with a high binding affinity [59]. SHP-1 is a non-receptor protein tyrosine phosphatase, which is predominantly expressed in hematopoietic cells and functions as a negative regulator of inflammation and tumor suppressor. Inhibition of SHP-1 activity resulted in promoted tumor growth and metastasis [123]. SHP-1 was also reported to be regulated by HNF1α in rat hepatocytes [124]. By interacting with SHP-1, HNF1α-AS1 increased the phosphatase activity of SHP-1 and performed its tumor-suppressing function.

Furthermore, HNF1α-AS1 was reported to interact with enzymes or complexes involved in epigenetic modifications. Using RNA immunoprecipitation, HNF1α-AS1 was found to be able to interact with DNA methyltransferase (cytosine-5) 1 (DNMT1) in A549 cell lines and polycomb repressive complex 2 member enhancer of zeste 2 (EZH2) complex in SMCC-7721 and Huh7 cell lines [91,92]. DNMT1 and EZH2 are critical proteins involved in DNA methylation and histone methylation, respectively. These findings suggest that HNF1α-AS1 might regulate target genes by altering their epigenetic markers through interaction with epigenetic modifying enzymes or complexes. The identified mechanisms of HNF1α-AS1 in gene regulation are summarized and displayed in Figure 9.

In term of HNF4α-AS1 in gene regulation, no clear mechanisms have been identified or studied. This question needs to be addressed by the future studies. 

## 6. Conclusions and Future Directions

Even though the study about lncRNAs is a relatively new research field, there is growing evidence showing the importance of lncRNAs to human health. Identifying and understanding disease or physiologically related lncRNAs have been a focus recently. However, the understanding of the roles of lncRNAs is still quite limited. Among all the 17,960 lncRNAs annotated and recorded in the GENCODE database, few have been functionally characterized [5]. Better understandings of the functions of each lncRNA are urgently needed to address the question on how knowledge about lncRNAs can benefit human health.

In the current review, the current findings of two lncRNAs, HNF1α-AS1 and HNF4α-AS1, are summarized and discussed. Several key features of these two lncRNAs have been identified in recent studies, including their involvement in human diseases and normal physiological activities.

In the human genome, HNF1α-AS1 and HNF4α-AS1 are antisense neighborhood lncRNAs to the TFs HNF1α and HNF4α, which are master regulators to several liver functions. HNF1α-AS1 and HNF4α-AS1 are found to be expressed in several organs in the human body, including the liver, kidney, GI tract, and pancreas in normal healthy conditions. Highly expressed HNF1α-AS1 and HNF4α-AS1 were also found in several cancerous tissues. Some of these cancers have an intrinsic expression of HNF1α-AS1 and HNF4α-AS1, such as colon cancer, HCC, or gastric cancer [59,83,85,98]. However, high levels of HNF1α-AS1 were also found in several cancers that have very low intrinsic HNF1α-AS1, such as lung cancer and cervical cancer [97,125]. This phenomenon suggested that dysfunction or dysregulation of lncRNAs is able to affect the health conditions of human beings.

HNF1α-AS1 and HNF4α-AS1 have been recently reported to regulate hepatic P450s in several *in vitro* hepatic cell models. These two lncRNAs showed opposite regulatory patterns toward DMEs and DME related TFs. Specifically, HNF1α-AS1 was responsible for the maintenance or upregulation of DMEs, while HNF4α-AS1 was able to repress DMEs. This two-way regulatory mechanism might be an explanation of how P450s respond to stimulates, where levels of P450s get induced and return to normal. However, in terms of how these two lncRNAs regulate their target genes is still not clear. The neighborhood TFs of HNF1α-AS1 and HNF4α-AS1, HNF1α and HNF4α, are well-studied master regulators of hepatic P450s. In this case, HNF1α-AS1 and HNF4α-AS1 might regulate P450s by affecting the functions of HNF1α and HNF4α, or independently.

HNF1α-AS1 and HNF4α-AS1 have also been found to highly involved in human diseases, especially cancer. Overexpressed HNF1α-AS1 has been found in a number of solid tumors. The expression level of HNF1α-AS1 was reported to correlate with several parameters in these cancers such as survival time, treatment outcome, and cancer risk. In most studies, a high expression level of HNF1α-AS1 was favorable to the progression of cancer. However, contradictory results also exist, adding the complexity of the HNF1α-AS1 functions in specific cases. HNF1α-AS1 was found to be involved in several important signaling pathways in cancer and inhibition of HNF1α-AS1 also showed to reduce the activities or spread cancer cells. This indicated that HNF1α-AS1 might have potential therapeutic benefits to cancer treatment. The study about HNF4α-AS1 is still very limited for now. The expression levels of HNF4α-AS1 was found to correlate with HCC and Crohn’s disease, making it a potential biomarker for the diagnosis of these diseases.

Several mechanism types of HNF1α-AS1 function have also been identified. Interacting with miRNAs and proteins have been the two major mechanism types in HNF1α-AS1 function. Other mechanisms are still waiting to be identified.

However, the research about HNF1α-AS1 and HNF4α-AS1 is still at the beginning and future studies are still needed to address the biological significance of these two lncRNAs. 

Firstly, according to the current studies, the function of these two lncRNAs can be greatly different based on the tissue or organ. In cancerous tissues, HNF1α-AS1 showed specific functions in cancer pathology and treatment based on the cancer type. However, in normal tissues, such as the liver, HNF1α-AS1 and HNF4α-AS1 had very different functions in physiological activities. Moreover, HNF1α-AS1 and HNF4α-AS1 are expressed in several different normal organs in the human, including the kidney and several GI track organs. Whether HNF1α-AS1 and HNF4α-AS1 have tissue-specific functions in these organs still needs to be addressed. 

Secondary, the potential therapeutic benefits of HNF1α-AS1 and HNF4α-AS1 need to be studied and uncovered in future studies. HNF1α-AS1 and HNF4α-AS1 are potential drug targets for future drug designs. HNF1α-AS1 and HNF4α-AS1 have been found to be highly involved in physiological and pathological activities in humans, making them great potential targets to treat diseases [126,127]. Conventional drugs and the recently developed siRNA-based drugs targeting HNF1α-AS1 and HNF4α-AS1, or other lncRNAs with potential therapeutic benefits, might be novel treatment strategies for human diseases in the future [128].

In conclusion, the current review summarizing the roles of HNF1α-AS1 and HNF4α-AS1 indicates that these two lncRNAs have critical roles in human diseases and physiological activities. However, the current research is incomplete in discovering the functions of HNF1α-AS1 and HNF4α-AS1 in human health and future studies are still needed.

## Figures and Tables

**Figure 1 ncrna-06-00024-f001:**
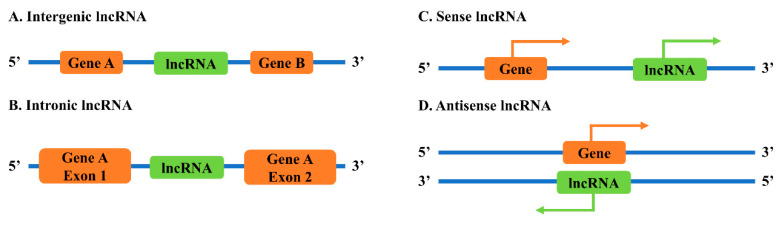
Classification of lncRNAs based on genomic locations with: (**A**) intergenic lncRNA; (**B**) intronic lncRNA; (**C**) sense lncRNA; and (**D**) antisense lncRNA.

**Figure 2 ncrna-06-00024-f002:**
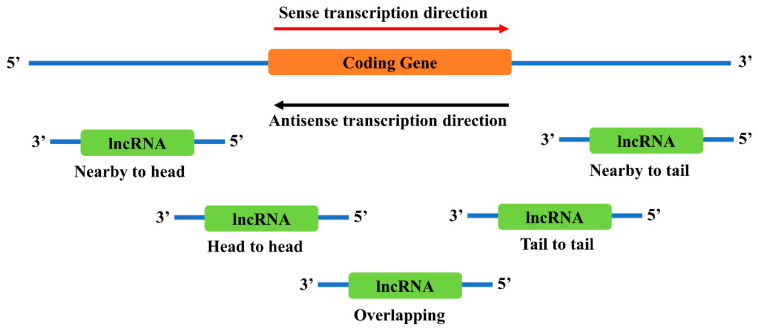
Genomic locations of antisense neighborhood lncRNAs to a sense coding gene. Depending on the relative locations towards the coding gene, neighborhood lncRNAs are categorized into several types: nearby to head, the 5’ end of the lncRNA is close to the 5’ end of the coding gene; head to head, the 5’ end of both genes are aligned together; overlapping, the antisense lncRNA overlaps with the sense coding gene; tail to tail, the 3’ end of both genes are aligned together; and nearby to tail, the 3’ end of the lncRNA is close to the 3’ end of the coding gene.

**Figure 3 ncrna-06-00024-f003:**
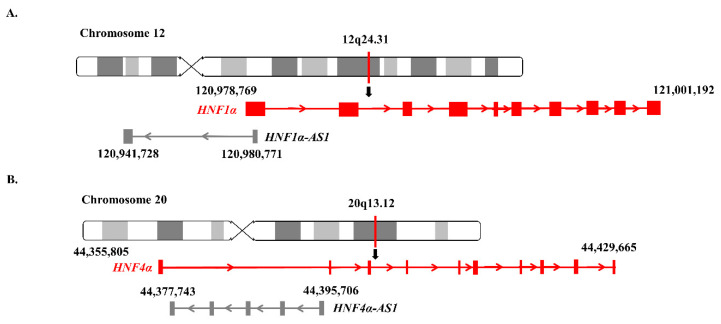
Genomic locations of *HNF1α, HNF1α-AS1, HNF4α,* and *HNF4α-AS1*: (**A**) locations of *HNF1α* and *HNF1α-AS1* on human chromosome 12; and (**B**) locations of *HNF4α* and *HNF4α-AS1* on human chromosome 20.

**Figure 4 ncrna-06-00024-f004:**
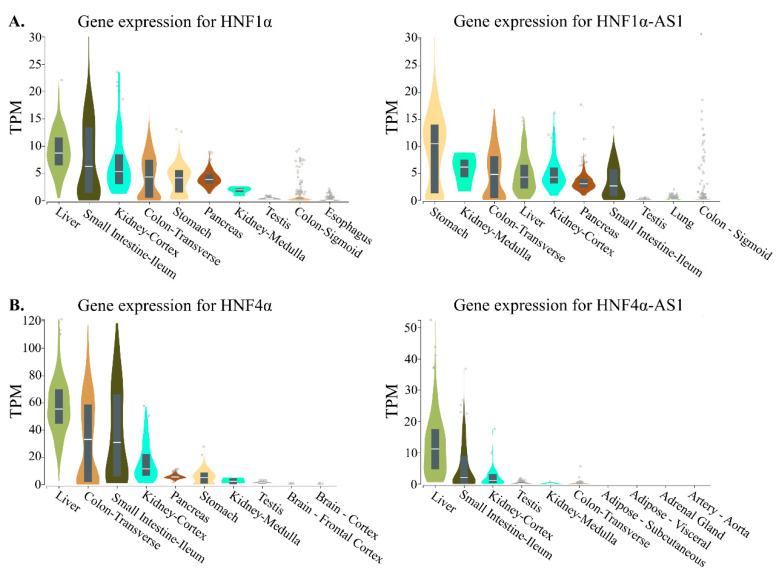
Top 10 tissues expressing HNF1α, HNF1α-AS1, HNF4α, and HNF4α-AS1 in human: (**A**) tissue expression patterns of HNF1α and HNF1α-AS1; and (**B**) tissue expression patterns of HNF4α and HNF4α-AS1. Data are from GTEx database (https://www.gtexportal.org/home/). TPM: transcripts per million.

**Figure 5 ncrna-06-00024-f005:**
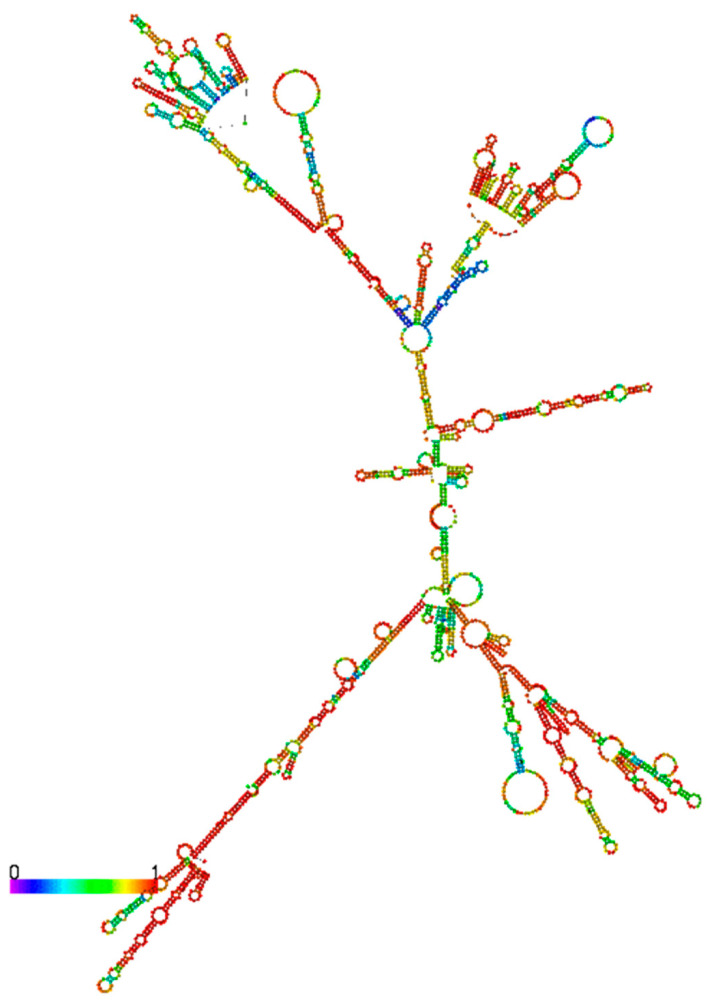
Predication of secondary structure of HNF1α-AS1 by RNAfold. The structure of HNF1α-AS1 is generated based on minimum free energy method. The colors indicate base-pair probabilities from low to high.

**Figure 6 ncrna-06-00024-f006:**
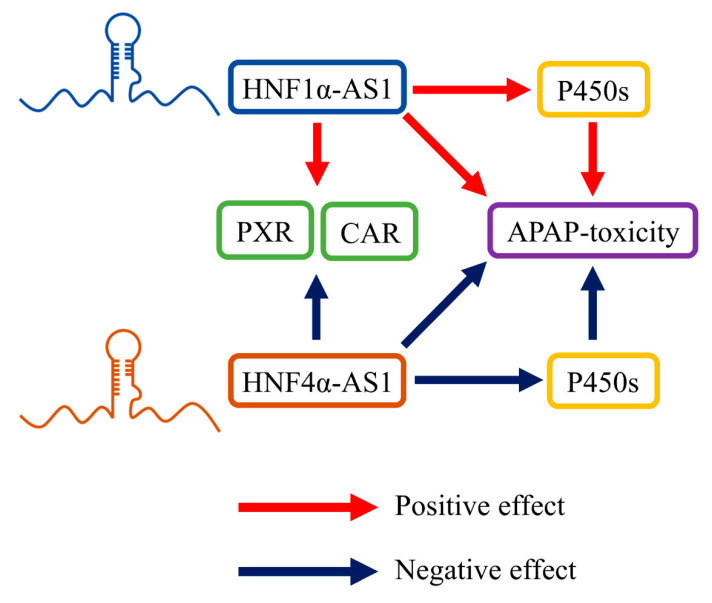
HNF1α-AS1 and HNF4α-AS1 in hepatic metabolism. HNF1α-AS1 is able to positively regulate the expression of PXR, CAR, and several P450s. HNF1α-AS1 also affected APAP-induced toxicity by regulating the expression and function of APAP-metabolizing P450s. Meanwhile, HNF4α-AS1 is able to negatively regulate the expression of PXR, CAR, P450s, and APAP-induced toxicity.

**Figure 7 ncrna-06-00024-f007:**
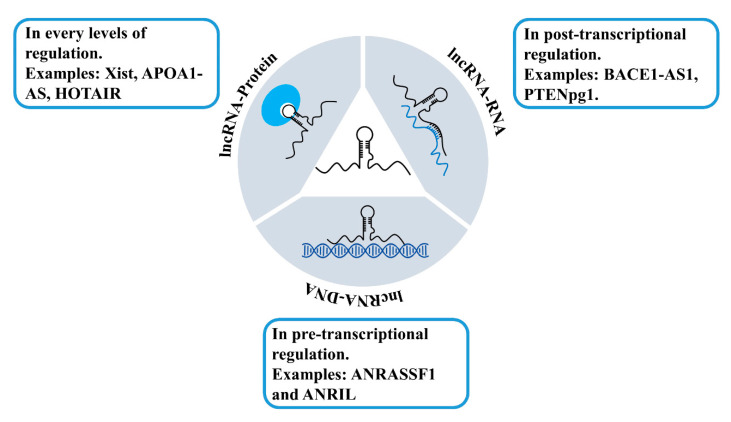
Examples of lncRNAs interacting with DNAs, RNAs, and proteins.

**Figure 8 ncrna-06-00024-f008:**
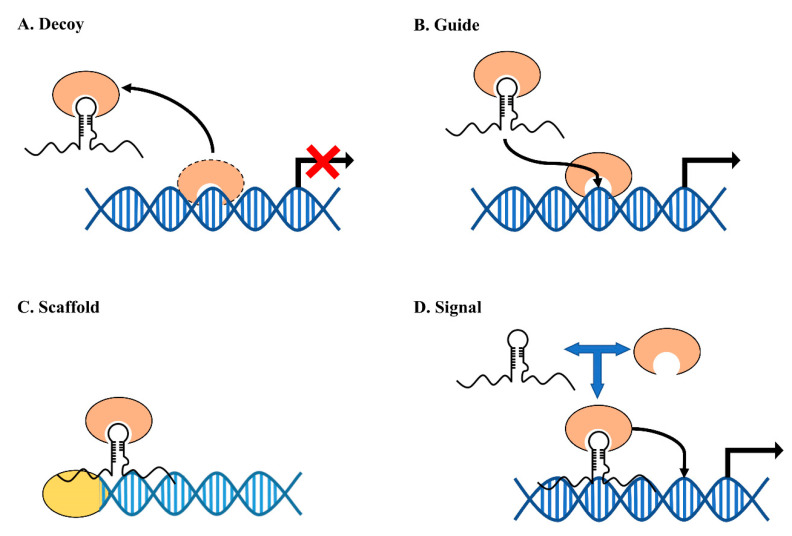
Schematic diagram of mechanisms of lncRNAs: (**A**) lncRNAs working in decoy mechanism can prevent the binding of a molecular to its target; (**B**) lncRNAs working in guide mechanism can direct the lncRNA-interacting molecules to specific genes or loci; (**C**) lncRNAs working in scaffold mechanism serve as central platforms for the binding of multiple molecules; and (**D**) lncRNAs working in signal mechanism can serve as ”switches” for the activation of specific genes.

**Figure 9 ncrna-06-00024-f009:**
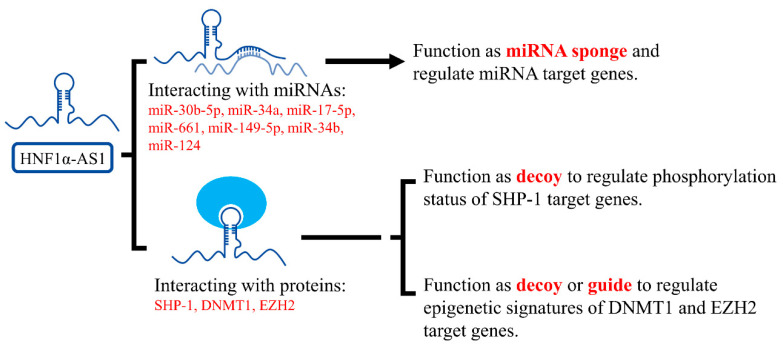
Mechanisms of HNF1α-AS1 in the regulation of gene expression. HNF1α-AS1 is currently found to interact with several miRNAs and proteins. By interacting with these molecules, HNF1α-AS1 can function through mechanisms of a miRNA sponge, decoy, or guide to regulate gene expression.

**Table 1 ncrna-06-00024-t001:** Summary of TF-neighborhood lncRNA pairs.

TF Gene	lncRNA Gene	Functions of TF Protein	Reference
*GATA3*	*GATA3-AS1*	Involve in differentiation and function of T helper 2 cells through regulation of GATA3, IL5, and IL13.	[32]
*GABPB1*	*GABPB1-AS1*	Regulate cellular antioxidant capacity and cell viability in HepG2 cells.	[33]
*YY1*	*lincYY1*	Promote myogenic differentiation and muscle regeneration.	[34]

**Table 2 ncrna-06-00024-t002:** The coding potentials of HNF1α-AS1 and HNF4α-AS1.

RNA	Fickett Score	Coding Probability	Classification
HNF1α	0.45633	1	Coding
HNF1α-AS1	0.28343	0.200677	Noncoding
HNF4α	0.34832	1	Coding
HNF4α-AS1	0.28415	0.353143	Noncoding

**Table 3 ncrna-06-00024-t003:** Summary of miRNAs interacting with HNF1α-AS1.

miRNA	Cancer Type	Reference
miR-30b-5p	HCC, bladder cancer	[84,89]
miR-34a	Colon cancer	[85]
miR-17-5p	Non-small cell lung cancer (NSCLC)	[93]
miR-661	Gastric cancer	[62]
miR-149-5p	NSCLC	[122]
miR-34b	Cervical cancer	[97]
miR-124	Colorectal cancer	[94]

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
