# Peer review of "The Roles of Long Noncoding RNAs HNF1α-AS1 and HNF4α-AS1 in Drug Metabolism and Human Diseases"

_ncrna, 2020, doi:10.3390/ncrna6020024_

Round 1

Reviewer 1 Report

This is an interesting, up-to-date, and well-developed review about HNF1a-AS1 and HNF4a-AS1, from a group with experience in studying these lncRNAs. However, the interest of the review would increase if the authors added information on orthologs in other species. Finally, just a little spelling mistake on line 396 ("cancellous tissues" should be "cancerous tissues").

Author Response

Reviewer 1

This is an interesting, up-to-date, and well-developed review about HNF1a-AS1 and HNF4a-AS1, from a group with experience in studying these lncRNAs. However, the interest of the review would increase if the authors added information on orthologs in other species. Finally, just a little spelling mistake on line 396 ("cancellous tissues" should be "cancerous tissues").

We appreciate the reviewer’s insightful comments and have corrected the spelling error on line 472 in the revision.

Reviewer 2 Report

In their manuscript, Chen & colleagues review the literature related to two antisense long non-coding RNAs, HNF1alpha-AS1 & HNF4alpha-AS1.

Overall, the review is quite clear and provides an appreciable global view of the current knowledge regarding these two non-coding RNAs. In this regard, it will probably interest the researchers working on these two transcripts. Perhaps the authors could try to emphasize to which extent and respect HNF1alpha-AS1 & HNF4alpha-AS1 could be used as paradigm to understand the function of antisense lncRNAs in human cells, in order to increase the impact of their review for a more general readership.

In addition, I think the manuscript needs to be improved before being accepted for publication. Here are my major suggestions, that I hope they will be useful and constructive.

1. To my opinion, there is an important work to do at the level of the figures. In fact, while Fig 1-2 provide informations directly related to the topic of the review, Fig 3-6 (as well as table 1) show generalities that can be found in many other reviews. They are therefore not very useful and appear to be out of the scope of this review. I would suggest to replace these « classical » figures by new, more specific figures directly illustrating some of the main messages of the manuscript. For example :

- one new figure illustrating the informations provided in paragraph 5.2 about the roles of HFN1/4alpha-AS1 and the effects of their know down

- another new figure illustrating the possible mechanisms of action of HFN1alpha-AS1 described in paragraph 6.2.

2. Consistent with major point 1, I suggest to restrict/shorten chapter 3 to the minimum and (perhaps) move the general informations about lncRNAs classification/location of antisense transcripts/examples of antisense controlling TFs into the introduction. Same for paragraph 6.1.

3. In paragraph 5.3., I would appreciate if the authors could provide informations (if available) about the role in cancer/disease of the « sense » genes related to the two antisense lncRNAs, ie HFN1a & HFN4a. It might be interesting to discuss as the authors mention that expression of HFN1a and the antisense lncRNA  are often positively correlated. Is it the same in cancers where the antisense has been described to be up- or down-regulated ?

Minor points :

1. line 34 : the fact that lncRNAs are devoid of coding potential is under debate, as at least a fraction of them have been shown to contain small ORFs bound by ribosomes, which sometimes gives rise to the production of peptides. This should be mentioned (for example, cite the latest study of the Weissman’s lab, PMID: 32139545).

2. line 40 : give some references for lncRNAs databases (ENCODE, MiTranscriptome, LNCipedia) and provide the exact current numbers of annotated transcripts.

3. lines 65-6 : present the results of coding prediction visuallly as a panel in Figure 1, not as a link.

4. Figure 2 : define « TPM » in the legends, and increase the font size for X-axis legends (as it is, it is impossible to read).

5. line 106  : explain the reasons of the statement that gene promoter binding cannot fully explain the functions of the two TFs.

6. Line 127 : cite more recent references for antisense annotation. Ref 20 was published in 15 years ago ! There are more recent works.

7. Line 230 : provide references.

8. line 266 :  I would not say that the mentioned results are controversial. It is just that the activity/effect of the transcript is probably tissue-specific.

9. line 273 : define HCC here (it is defined after, in line 362).

10. line 352 : perhaps HFN4alpha-AS1 could be removed from the title as no information is available for it.

11. line 385 : give exact numbers, referring to the latest version of lncRNA databases (LNCpedia, etc - see minor point 2).

12. The conclusion could be shortened. Avoid redundancy with the main text, rather go directly to the main conclusions and perspectives.

Author Response

Reviewer 2

In their manuscript, Chen & colleagues review the literature related to two antisense long noncoding RNAs, HNF1a-AS1 and HNF4a-AS1.

Overall, the review is quite clear and provides an appreciable global view of the current knowledge regarding these two noncoding RNAs. In this regard, it will probably interest the researchers working on these two transcripts. Perhaps the authors could try to emphasize to which extent and respect HNF1a-AS1 and HNF4a-AS1 could be used as paradigm to understand the function of antisense lncRNAs in human cells, in order to increase the impact of their review for a more general readership.

In addition, I think the manuscript needs to be improved before being accepted for publication. Here are my major suggestions, that I hope they will be useful and constructive. 

  1. To my opinion, there is an important work to do at the level of the figures. In fact, while Fig 1-2 provide information directly related to the topic of the review, Fig 3-6 (as well as table 1) show generalities that can be found in many other reviews. They are therefore not very useful and appear to be out of the scope of this review. I would suggest to replace these « classical » figures by new, more specific figures directly illustrating some of the main messages of the manuscript. For example:

- one new figure illustrating the information provided in paragraph 5.2 about the roles of HFN1/4a-AS1 and the effects of their know down

- another new figure illustrating the possible mechanisms of action of HFN1a-AS1 described in paragraph 6.2.

We appreciate the reviewer’s constructive comments and have added two new figures (Figure 6 in line 311 and Figure 9 in line 451 in these sections to summarized the information about HNF1a-AS1 and HNF4a-As1’s functions and mechanisms.

  1. Consistent with major point 1, I suggest to restrict/shorten chapter 3 to the minimum and (perhaps) move the general information about lncRNAs classification/location of antisense transcripts/examples of antisense controlling TFs into the introduction. Same for paragraph 6.1.

We appreciate the reviewer’s comment. However, putting all the introductory sections together will make the introduction too long and the review into two separate parts. In this case, we insisted to separate the introduction part and incorporate the information into each section when we discuss our target lncRNAs. We hope the reviewer accept our explanation.

  1. In paragraph 5.3., I would appreciate if the authors could provide information (if available) about the role in cancer/disease of the sense genes related to the two antisense lncRNAs, ie HFN1a & HFN4a. It might be interesting to discuss as the authors mention that expression of HFN1a and the antisense lncRNA are often positively correlated. Is it the same in cancers where the antisense has been described to be up- or down-regulated? 

We would like to appreciate the reviewer’s helpful suggestion. Some additional discussion about roles of HFN1a & HFN4a in cancer has been added to the corresponding sections (line 219-230). We agree with the idea that it will be interesting to compare the expression of the coding and noncoding genes in the same types of cancer. However, in most references, the authors did not show this part of data, so we can’t study the correlation between the coding and noncoding genes. Also, we tend to believe that at least part of the functions of lncRNAs in cancer are independent of their coding genes, as some of the coding genes are not reported to play a role in the specific cancer types.

Minor points:

  1. Line 34: the fact that lncRNAs are devoid of coding potential is under debate, as at least a fraction of them have been shown to contain small ORFs bound by ribosomes, which sometimes gives rise to the production of peptides. This should be mentioned (for example, cite the latest study of the Weissman’s lab, PMID: 32139545).

We have followed the comment and added the related information in line 32-39 with citation of reference [3] in the revision.

  1. Line 40: give some references for lncRNAs databases (ENCODE, MiTranscriptome, LNCipedia) and provide the exact current numbers of annotated transcripts.

We appreciate the reviewer’s comment. Several lncRNA database have been added in the revision in line 43-45.

  1. Lines 65-6: present the results of coding prediction visually as a panel in Figure 1, not as a link.

We appreciate the reviewer’s comment and have added this information in the revision.

  1. Figure 2: define TPM in the legends and increase the font size for X-axis legends (as it is, it is impossible to read).

We have modified the figure to make it easier to read (new Figure 4 line 181). TPM is defined in the figure legend.

  1. Line 106: explain the reasons of the statement that gene promoter binding cannot fully explain the functions of the two TFs.

We appreciate the reviewer’s comment and have modified the statement to make it more clear in the revised manuscript (line 235-236).

  1. Line 127: cite more recent references for antisense annotation. Ref 20 was published in 15 years ago! There are more recent works.

We appreciate the reviewer’s insightful comment and have added some more recent references about antisense transcripts (line 81-84). However, we kept the reference published on Science on 2005 as it was still one of the most significant paper talking about antisense transcripts in human genome.

  1. Line 230: provide references.

We have added the related references following the sentence.

  1. Line 266: I would not say that the mentioned results are controversial. It is just that the activity/effect of the transcript is probably tissue-specific.

We appreciate the reviewer’s constructive comment and have corrected the statement in the revised manuscript (line 333-334).

  1. Line 273: define HCC here (it is defined after, in line 362).

We appreciate the reviewer’s comment. HCC is first mentioned in line 131 in the revision, where the full name is mentioned. We have removed the full name in line 362 to avoid confusion.

  1. Line 352: perhaps HFN4a-AS1 could be removed from the title as no information is available for it.

We appreciate the reviewer’s comment and have applied this change in the revision (line 421).

  1. Line 385: give exact numbers, referring to the latest version of lncRNA databases (LNCpedia, etc - see minor point 2).

We appreciate the reviewer’s comment and have added the exact number of annotated lncRNAs according to the GENCODE database (line 461).

  1. The conclusion could be shortened. Avoid redundancy with the main text, rather go directly to the main conclusions and perspectives.

We appreciate the reviewer’s comment. The last section of the review is consisted with both conclusion and future directions about lncRNA research.

Reviewer 3 Report

Chen and co-authors review the data about functionallity of lncRNAs HNF1α-AS1 and HNF4α-AS1, an antisense lncRNAs encoded near to transcription factors HNF1α-AS1 and HNF4α-AS1 transcripts, respectively, focusing in cellular metabolism during physiological and pathological conditions.Authors carefully revised the genomic location, transcription and interactions of lncRNAs with TF. Also, explored the literature about lcnRNAs-TF interactions and functions in gene expression regulation, as well as miRNAs sponge. I recommend the publication of article after minor revision.

Poin-by-point:

Introduction,  section 1:

Re-write the phrase  to turn the sentence clear: "Not only the coding genes, which only takes up a small portion in the human genome, but also the noncoding genes have been investigated in recent studies.".

Section 2, in Figure 2, enlarge fonts from X-axis.

Section 2, I suggest to include the prediction of molecular structure of lncRNA and putative binding sites of TF and lncRNAs.

The 2 paragraphs in lines 84-109 from section 2 (genomics location and structure) can be better merged with section 4, because explain the molecules functions and involvement in diseases.

In section 3, what is the reference for phrase:"Recent studies have found that these neighborhood lncRNAs might affect their nearby coding genes in multiple ways, including regulation of the coding genes or involvement in the functional regulation of the coding genes."?

In section 3.2: Authors can include some works showing that similar to lncRNAs and mRNAs, antisense transcripts may be capped and poly-adenylated and are maturated to excise introns. Also, their transcription can be controlled by promoters and enhancers, as well as discussions about bidirectional promoters that control transcript expression originating from both strands (References: doi: 10.1126/science.1162253; doi: 10.3390/ijms17010009; doi: 10.1016/j.gene.2005.10.009.; doi: 10.18632/oncotarget.6255)

Full name of PRDX5, line 160.

Include short name EGR1 for Early growth response protein 1, line 190.

Authors can include the impact of miRNA sponge activity of HNF1α-AS1in  the section 6.2.

Author Response

Reviewer 3

Chen and co-authors review the data about functionality of lncRNAs HNF1α-AS1 and HNF4α-AS1, an antisense lncRNAs encoded near to transcription factors HNF1α-AS1 and HNF4α-AS1 transcripts, respectively, focusing in cellular metabolism during physiological and pathological conditions. Authors carefully revised the genomic location, transcription and interactions of lncRNAs with TF. Also, explored the literature about lncRNAs-TF interactions and functions in gene expression regulation, as well as miRNAs sponge. I recommend the publication of article after minor revision.

Point-by-point:

Introduction, section 1: Re-write the phrase to turn the sentence clear: "Not only the coding genes, which only takes up a small portion in the human genome, but also the noncoding genes have been investigated in recent studies." 

We appreciate the reviewer’s comment and have rewrite the sentence to make it clear.

Section 2, in Figure 2, enlarge fonts from X-axis.

We appreciate the reviewer’s comment and have modified new Figure 4.

Section 2, I suggest to include the prediction of molecular structure of lncRNA and putative binding sites of TF and lncRNAs.

We appreciate the reviewer’s comment and have added some discussion about the structure in the corresponding section.

The 2 paragraphs in lines 84-109 from section 2 (genomics location and structure) can be better merged with section 4, because explain the molecules functions and involvement in diseases.

We appreciate the reviewer’s comment. However, the genomic locations and tissue distribution of lncRNAs cannot fully explain their functions. To avoid the misleading, we separate these two sections.

In section 3, what is the reference for phrase: "Recent studies have found that these neighborhood lncRNAs might affect their nearby coding genes in multiple ways, including regulation of the coding genes or involvement in the functional regulation of the coding genes."?

We appreciate the reviewer’s comment and have added some reference talking about how neighborhood lncRNAs affecting nearby coding genes.

In section 3.2: Authors can include some works showing that similar to lncRNAs and mRNAs, antisense transcripts may be capped and poly-adenylated and are maturated to excise introns. Also, their transcription can be controlled by promoters and enhancers, as well as discussions about bidirectional promoters that control transcript expression originating from both strands (References: doi: 10.1126/science.1162253; doi: 10.3390/ijms17010009; doi: 10.1016/j.gene.2005.10.009.; doi: 10.18632/oncotarget.6255)

We appreciate the reviewer’s comment. We have added related discussion about this in the current section in the revision.

Full name of PRDX5, line 160.

We have added the full name of PRDX5.

Include short name EGR1 for Early growth response protein 1, line 190.

We appreciate the reviewer’s comment. However, the name “early growth response protein 1” only appeared once in the manuscript, so we did not include the abbreviation for it.

Authors can include the impact of miRNA sponge activity of HNF1α-AS1in the section 6.2.

We appreciate the reviewer’s comment. By interacting with miRNAs, HNF1α-AS1 was able to disinhibit these miRNA target genes. Based on the interacting miRNAs, this interaction can affect different genes involved in all the activities we mentioned in section 5.1 (original section 6.1). We did not mention this in section 5.2 (original section 6.2) to avoid repeating the information.

Reviewer 4 Report

The manuscript by Chen et al. is a review summarizing the current knowledge on the two long non-coding RNAs (lncRNAs) HNF1α-AS1 and HNF4α-AS1.

The authors try to elucidate the importance of these lncRNAs and their involvement in a diversity of human physiological activities and diseases.

They explain typical characteristics of lncRNAs exemplary on well-studied lncRNAs and show how these characteristics apply to HNF1α-AS1 and HNF4α-AS1.

HNF1α-AS1 and HNF4α-AS1 are described as typical lncRNAs with low coding potential and a genomic location antisense to the locus of their corresponding genes HNF1α and HNF4α. The lncRNAs share similar organ-specific expression patterns as their corresponding genes (DOI: 10.1038/ng.2653). The characteristics of neighbourhood antisense lncRNAs of transcription factors (TFs) is described and how this applies for both lncRNAs anf corresponding TFs (DOI: 10.1038/nrg3594). Both lncRNAs are described to be involved in drug metabolism and a variety of cancers through altering gene expression (DOI: 10.1124/mol.119.118778). HNF1α-AS1 is said to execute these functions through miRNA sponging and interaction with epigenetic modifying proteins (doi: 10.18632/oncotarget.3247). For HNF4α-AS1 the mechanisms are largely unknown.

Therefore, the reviewer has the following points:

  1. Abstract: Functions and involvement of HNF1α-AS1 and HNF4α-AS1 could be more specific in the abstract, e.g. about the interaction partners and physiological activities, diseases and types of cancer HNF1α-AS1 and HNF4α-AS1 interact with and are involved in.
  2. The authors illustrate important features and characteristic of lncRNAs in detail. This helps to explain the features observed in HNF1α-AS1 and HNF4α-AS1.

This exemplary description of known lncRNAs should be shortened as the focus of this review is said to be on HNF1α-AS1 and HNF4α-AS1 and not other lncRNAs.

This applies especially to sections 3.3 “Neighbourhood antisense lncRNAs of TFs” and 6.1 “lncRNAs in gene regulation: functions and mechanisms”

  1. Section 2.; Figure 1: There is no information on the isoforms and length of the transcripts of HNF1α-AS1 and HNF4α-AS1. The genomic locus of both lncRNAs is explained correctly. It is not mentioned that HNF1α-AS1 has seven isoforms and HNF4α-AS1 has two isoforms. These isoforms vary in size, exon count and potentially tissue specific expression. Please specify this.
  2. The entire third section only provides background information and examples of functions of lncRNAs corresponding to their genomic locus and neighbourhood. It would make sense to insert these information into section 2, where the locations and genomic neighbourhood of HNF1α-AS1 and HNF4α-AS1 are presented.
  3. It could help the logical structure of the review to specifiy more on potential tissue-specific expression of HNF1α-AS1 and HNF4α-AS1 and their corresponding proteins.
  4. The coding potential calculator is a usual tool to predict coding potential of lncRNAs. It seems as though the genomic sequence was used for the prediction. Optimally, the sequence of the predominant isoform of the lncRNA is used. This can provide more specificity in the prediction
  5. Section 3: Here the distinction between cis- and trans-acting lncRNAs should be added. The authors do not mention this important functional feature of lncRNAs.
  6. Figure 1: The way the names of the genes and lncRNAs are sown can be misleading. The Boxes around the names are attached to each respective first exon. Therefore, the first exon appears to be larger than its actual size.
  7. Figure 2: Please explain the abbreviation of the Y-axis labelling (TPM).
  8. Section 5.2 (lines 251f): The authors should provide concrete examples of other lncRNAs that have been identified to be involved in the regulation of expression of drug metabolizing enzyme genes and contribution to susceptibility of drug-induced liver injury (DILI).
  9. On several occasions the authors mention “normal tissue” (line 69, 263, 428), “normal organs” (line 430) and “normal physiological activities”. It would be worth considering changing these to terms that are more specific.
  10. Conclusion: The authors only summarize the presented data. Please provide a directive for future research on these lncRNAs.

Author Response

Reviewer 4

The manuscript by Chen et al. is a review summarizing the current knowledge on the two long noncoding RNAs (lncRNAs) HNF1α-AS1 and HNF4α-AS1.

The authors try to elucidate the importance of these lncRNAs and their involvement in a diversity of human physiological activities and diseases.

They explain typical characteristics of lncRNAs exemplary on well-studied lncRNAs and show how these characteristics apply to HNF1α-AS1 and HNF4α-AS1.

HNF1α-AS1 and HNF4α-AS1 are described as typical lncRNAs with low coding potential and a genomic location antisense to the locus of their corresponding genes HNF1α and HNF4α. The lncRNAs share similar organ-specific expression patterns as their corresponding genes (DOI: 10.1038/ng.2653). The characteristics of neighborhood antisense lncRNAs of transcription factors (TFs) is described and how this applies for both lncRNAs and corresponding TFs (DOI: 10.1038/nrg3594). Both lncRNAs are described to be involved in drug metabolism and a variety of cancers through altering gene expression (DOI: 10.1124/mol.119.118778). HNF1α-AS1 is said to execute these functions through miRNA sponging and interaction with epigenetic modifying proteins (doi: 10.18632/oncotarget.3247). For HNF4α-AS1 the mechanisms are largely unknown.

Therefore, the reviewer has the following points:

  1. Abstract: Functions and involvement of HNF1α-AS1 and HNF4α-AS1 could be more specific in the abstract, e.g. about the interaction partners and physiological activities, diseases and types of cancer HNF1α-AS1 and HNF4α-AS1 interact with and are involved in.

We appreciate the reviewer’s comment and have modified the abstract in the revision. However, the journal has a limitation of 200 words in the abstract, so we cannot go too detail about the results.

  1. The authors illustrate important features and characteristic of lncRNAs in detail. This helps to explain the features observed in HNF1α-AS1 and HNF4α-AS1. This exemplary description of known lncRNAs should be shortened as the focus of this review is said to be on HNF1α-AS1 and HNF4α-AS1 and not other lncRNAs. This applies especially to sections 3.3 “Neighborhood antisense lncRNAs of TFs” and 6.1 “lncRNAs in gene regulation: functions and mechanisms”

We appreciate the reviewer’s insightful comment. We have kept the information for other lncRNAs very briefly.

  1. Section 2.; Figure 1: There is no information on the isoforms and length of the transcripts of HNF1α-AS1 and HNF4α-AS1. The genomic locus of both lncRNAs is explained correctly. It is not mentioned that HNF1α-AS1 has seven isoforms and HNF4α-AS1 has two isoforms. These isoforms vary in size, exon count and potentially tissue specific expression. Please specify this.

We appreciate the reviewer’s constructive comment. The information showed on Figure 1 (new Figure 3) is incorporated from the UCSC genome browser. Length of HNF1α-AS1 and HNF4α-AS1 can be calculated based on the start and end position of the gene. Various transcript isoforms of both lncRNAs are annotated but have not been confirmed experimentally yet. Whether different isoforms of these two lncRNAs have different functions are still largely unknown. In this case, we don’t discuss the isoforms of these two lncRNAs in the current review.

  1. The entire third section only provides background information and examples of functions of lncRNAs corresponding to their genomic locus and neighborhood. It would make sense to insert this information into section 2, where the locations and genomic neighborhood of HNF1α-AS1 and HNF4α-AS1 are presented. It could help the logical structure of the review to specify more on potential tissue-specific expression of HNF1α-AS1 and HNF4α-AS1 and their corresponding proteins.

We appreciate the reviewer’s comment and have modified the content based on the input of the comment in the revision.

  1. The coding potential calculator is a usual tool to predict coding potential of lncRNAs. It seems as though the genomic sequence was used for the prediction. Optimally, the sequence of the predominant isoform of the lncRNA is used. This can provide more specificity in the prediction

We appreciate the reviewer’s comment. The RNA sequences used for coding potential prediction were obtained from NCBI database, where only one reference sequence was given for both HNF1α-AS1 and HNF4α-AS1, which were shown as VALIDATED on the website. (https://www.ncbi.nlm.nih.gov/gene/101927219#reference-sequences , https://www.ncbi.nlm.nih.gov/gene/283460#reference-sequences )

  1. Section 3: Here the distinction between cis- and trans-acting lncRNAs should be added. The authors do not mention this important functional feature of lncRNAs.

We appreciate the reviewer’s comment and have added some introduction about the cis- and trans- acting lncRNAs in section 3.1.

  1. Figure 1: The way the names of the genes and lncRNAs are sown can be misleading. The Boxes around the names are attached to each respective first exon. Therefore, the first exon appears to be larger than its actual size.

We appreciate the reviewer’s comment and have modified new Figure 3 to make this clear.

  1. Figure 2: Please explain the abbreviation of the Y-axis labelling (TPM).

We appreciate the reviewer’s comment and have explained the TPM in the figure legend of new Figure 4.

  1. Section 5.2 (lines 251f): The authors should provide concrete examples of other lncRNAs that have been identified to be involved in the regulation of expression of drug metabolizing enzyme genes and contribution to susceptibility of drug-induced liver injury (DILI).

We appreciate the reviewer’s comment. In the manuscript, I added two reference talking about two other lncRNAs, such as LINC00574 and LINC00844, which were reported to involve in regulation of DME and susceptibility of DILI.

  1. On several occasions the authors mention “normal tissue” (line 69, 263, 428), “normal organs” (line 430) and “normal physiological activities”. It would be worth considering changing these to terms that are more specific.

We appreciate the reviewer’s comment. In line 69, the “normal human tissues” refers to all recorded human tissues in the database. In this case, it would be not necessary to list all these tissues. We have added some specific explanations in the other places. 

  1. Conclusion: The authors only summarize the presented data. Please provide a directive for future research on these lncRNAs.

We appreciate the reviewer’s comment. We have added related discussion about the future directions on lncRNA research in the last section of the revision.

Round 2

Reviewer 2 Report

Although the authors did not follow all my suggestions (which were only suggestions), I appreciate the modifications and new figures in this revised version of the review. I am happy to recommend the manuscript to be accepted for publication.